# Medical Education: Patients’ Perspectives on Clinical Training and Informed Consent

**DOI:** 10.3390/ijerph19137611

**Published:** 2022-06-22

**Authors:** Inês Gil-Santos, Cristina Costa Santos, Ivone Duarte

**Affiliations:** 1Faculty of Medicine, University of Porto, 4200-319 Porto, Portugal; ineesgildossantos@gmail.com; 2Department of Obstetrics and Gynecology, Centro Materno Infantil do Norte, Centro Hospitalar Universitário do Porto, 4099-001 Porto, Portugal; 3Department of Community Medicine, Information and Health Decision Sciences (MEDCIDS), Faculty of Medicine, University of Porto, 4200-319 Porto, Portugal; csantos@med.up.pt; 4Center for Health Technology and Services Research (CINTESIS), Faculty of Medicine, University of Porto, 4200-319 Porto, Portugal

**Keywords:** medical education, medical ethics, autonomy, patients’ perspectives

## Abstract

There are complex ethical dilemmas inherent to medicine teaching, particularly in clinical practice involving actual patients. Questions must be raised on fulfilling medical students’ training needs while still respecting patients’ fundamental rights to autonomy and privacy. We aimed to assess patients’ perspectives regarding medical students’ involvement in their medical care. An observational, cross-sectional study was developed, and a questionnaire was applied randomly to patients waiting for a consultation/admitted to three distinct departments: General Surgery, Obstetrics/Gynaecology, and Infectious Diseases. Of the 77% interviewed patients who reported previous experiences with medical students, only 59% stated that they were asked for consent for their participation, and 28% stated that students had adequately introduced themselves. Patients from Gynaecology/Obstetrics were the ones who reported lower rates of these practices and were also the ones who were most bothered by students’ presence, stating that they would be more comfortable without the presence of medical students. Male patients received more explanations than female patients regarding the same matters. Thirty-five percent of patients stated that they would feel more comfortable without the medical students’ presence. The study shows a need to pay closer attention to fulfilling patients’ fundamental rights.

## 1. Introduction

It is well established that bedside teaching has numerous benefits in medical education as it provides a unique learning opportunity for acquiring clinical knowledge, developing clinical skills, communication, and other soft skills, teaching empathy and facilitating the patient-doctor relationship [1,2,3,4].

Teaching constitutes an essential part of a doctor’s education, and is widely accepted by medical students, trainees and doctors [5,6], but it poses a series of ethical dilemmas and professional demands that must be carefully identified and adequately addressed. Even so, the medical education literature continues to lack substance relating to ethics in medical education, especially on medical clinical training involving actual patients and the ethical issues involved [5,7].

There has been a constant concern as to how to correctly approach and respond to these quandaries, and the main questions have been the protection of patients’ privacy and confidentiality and the practice of an ethical medicine and medical education. The once sovereign paternalistic model of medicine excluded the revealing of all information and precluded informed patient consent. The patient was expected to unquestionably trust their doctor, with no regard to their own preferences [7]. However, in an era where patient-centred medicine has become a staple, the principle of autonomy, i.e., the right to self-determination, has become the central ethical principle in healthcare [8,9].

As important as it may be, medical students’ presence is not essential to providing patients with medical assistance, and it contradicts their expectations to be seen by actual practicing doctors. When attending a teaching hospital in Portugal, it is expected that patients know there is a possibility they’ll be asked if they will allow medical students to participate in their medical care [10]. Portuguese law states that every intervention relating to someone’s health should only be performed after consent [11] and codes of conduct and ethical practice specifically designed for medical students exist. However, not all patients are fully aware of their rights, and proper consent is not always pursued [12]. Healthcare professionals are required to provide complete information to the patients and to pursue a process of active communication between them, which cannot be completed in a single conversation and often results in the patient being put on the spot in front of the students and therefore feeling obligated to consent [13].

Despite the growing concern with complementing medical students’ programmes in the medical ethics area [14,15], studies have pointed out that students tend to become less and less affected by the ethical dilemmas they face throughout their medical education. Basic ethical norms such as introducing themselves to the patients as students or requesting patients’ consent before history taking or physical examination are forgotten or ignored [16].

Previous studies have pointed out that one of the biggest motivators for patients’ acceptance of students’ participation in their medical care is an aspiration to contribute to their education, and the most common reason for refusal is concern about their privacy not being respected. [8] However, most patients show a positive attitude towards medical students’ participation as long as they are informed of the students’ presence and asked for their consent beforehand. The medical specialty in question has also been shown to influence findings; for instance, results have pointed out that obstetrics and gynaecology and surgery patients are less likely to accept students than patients in general practice [17]. Despite this, some studies point out that even in contexts such as labour and delivery or during a gynaecological examination, women were supportive of student participation if they felt they were being respected and their confidentiality protected [1].

That being said, questions must be raised as to how to fulfil medical students’ training needs as future doctors while still respecting patients’ fundamental rights to autonomy and privacy [8]. These issues are particularly pressing when it comes to invasive, uncomfortable, and intimate procedures such as gynaecological examination or invasive procedures, as patients are more inclined to refuse the presence/participation of medical students [18].

When faced with this ethical dilemma, the questions we pose are: what are patients’ perspectives on their involvement in medical education? Is the presence of medical students a stress factor for patients, or do they feel integrated and willing to participate in their medical education? Is their medical care affected? Are their rights being disrespected? This study intended to answer these questions by evaluating students’ presence and participation’s effect on patients’ quality of care, from their point of view. We aimed to assess whether there was the need to implement further measures to regulate student-patient interactions or intervene at an earlier stage of their medical education.

## 2. Materials and Methods

### 2.1. Study Design and Participants

This study was conducted over one month at a central University Hospital in Porto, Portugal, the largest hospital unit in the north region and one of the biggest in the country. It constitutes a highly specialised reference centre, counting 1062 acute care beds, 43 cribs, and 14 beds for admission to the Physical Rehabilitation department, 32 operating rooms, five birthing rooms, 252 consulting rooms and 135 chairs or beds. Every day about 15,000 to 20,000 patients visit its facilities. An observational, cross-sectional study was developed, and a questionnaire was applied. The data collection period was between January 2019 and March 2019.

The ethical procedures followed the Helsinki Declaration, and the Ethics Committee of the São João hospital complex analysed and approved the study (ref.366/18). All participants gave their informed consent.

The sample consisted of 131 patients interviewed in three different hospital departments: General Surgery, Infectious Diseases and Obstetrics/Gynaecology. Most patients were interviewed while waiting for a scheduled consultation, and 39% were interviewed while admitted to one of the previously mentioned departments. Interviewed patients were an average age of 49 years old. Of the 131 patients, more than half were female, and most patients were married. Only 21% of patients had completed a higher education (equivalent to a college degree) (Table 1).

### 2.2. Survey Questionnaire

A questionnaire (Appendix A) was administered randomly to patients waiting for a scheduled consultation or admitted to three distinct departments: General Surgery, Obstetrics/Gynaecology, and Infectious Diseases. The same investigator conducted all interviews. A total of 131 patients were interviewed after being appropriately informed of the study’s objectives and contents and giving consent. Minors (less than 18 years of age) and/or individuals unable to provide consent were excluded from the study. Patients were asked to supply demographic information, i.e., age, civil status and education level, and to answer some specific questions, i.e., whether a medical student had ever been present during their previous visits to health care institutions, followed by a series of closed—yes or no—questions on their perceptions of medical students’ involvement in their health care. Questions were constructed based on the most common issues found in the literature. Only patients with past experiences involving medical students answered the totality of the questions, given that the others provided only demographic information.

### 2.3. Statistical Analysis

The Chi-Square Test was used to compare patients’ responses according to the three different departments where interviews were conducted and patients’ level of education. The mean age of patients who answered affirmatively and negatively to each question was analysed using an Independent Sample T-Test.

A database was created based on the obtained answers, and data was analysed using the Statistical Package for Social Sciences (SPSS). A *p*-value < 0.05 was considered the cut-off value for statistical significance.

## 3. Results

Of the 131 interviewed patients, only 101 (77%) reported previous experiences where medical students had been present in a consultation and/or hospital admission context. Patients reported a median of three medical students present, with a minimum of 1 and a maximum of 12 students. None of the participants reported any situation where a student had been disrespectful towards them.

Only 59% of the patients with previous experiences regarding medical students’ involvement in their medical care reported that the physician asked them if the students could be present, and only 28% stated that the doctor introduced the students by name and with their year of medical training. The department from which the interviewed patients came was significantly associated with the patients’ responses, being that patients interviewed in the Obstetrics/Gynaecology department were the ones who reported a lower incidence of these practices. In contrast, patients interviewed in the Infectious Diseases department had a higher rate of affirmative answers to these questions (*p* = 0.006 and *p* = 0.001, respectively) (Table 2).

About a fifth of the patients stated that they felt uncomfortable with the students’ presence, and the department from which the interviewed patients came was significantly associated with this feeling (*p* = 0.002), since patients who primarily objected were from Obstetrics/Gynaecology (Table 2).

Only about half of the patients stated that the medical students introduced themselves as students and asked for their consent previously to taking their history or to a physical examination. A similar percentage said that they explained the procedures they were going to perform while addressing patients’ doubts and questions. The department in which patients gave a lower rate of affirmative answers to these questions was Obstetrics/Gynaecology, with only 30% of patients stating that the medical students introduced themselves and asked for consent and 27% reporting that they explained what they wanted to do and clarified patients’ doubts (*p* = 0.004 and *p* < 0.001, respectively). The department with the most affirmative answers was Infectious Diseases.

Forty-one per cent of the interviewed individuals said that if the problem in question-related to a more intimate part of their body, they would feel more bothered by the presence of medical students. Once again, the department from which the interviewed patients came was significantly associated with patients’ responses (*p* = 0.008), and the department in which patients expressed this feeling the most was also Obstetrics/Gynaecology (Table 2).

Thirty-five per cent of patients declared that they would feel more comfortable without the presence of medical students. This percentage was the highest amongst Obstetrics/Gynaecology patients (53%) and the lowest among those from the Infectious Diseases department (19%).

Table 3 describes the answers given by the interviewed patients according to their level of education (years of school attendance), which revealed that there was only a significant association with the difficulty of disclosure of an intimate problem by the patients in the presence of medical students. Seventy-one percent of the patients with a higher education stated that it would be more difficult for them to reveal/talk about an intimate problem in the presence of students, while only 28% of patients who completed four years or less of school and 27% of those who completed between 5 and 12 years of school responded affirmatively to that question (*p* = 0.001).

Patients’ acceptance of and perspectives regarding students’ involvement was also found to be significantly associated with their age. Patients who felt discomfort with the students’ presence were significantly younger than those who didn’t (*p* = 0.006). Younger patients were also the ones who said they would feel more bothered by the students’ presence in case the problem in question involved a more intimate part of their bodies (*p* = 0.014). On the other hand, patients who reported having been asked for consent by students who introduced themselves as such were significantly older than those who stated otherwise (*p* = 0.017) (Table 4).

Given that only female patients were interviewed from the Obstetrics/Gynaecology department and patients interviewed from the Infectious Diseases department were mainly male, a gender comparison was only possible between patients from the General Surgery department. There was a significant difference between students’ conduct regarding male and female patients; namely, when present in a setting of consultation or hospital admission, students explained the intended procedures and clarified patients’ doubts to more than half of the male patients (56%), but that was only the case for less than a quarter (19%) of the female patients (*p* = 0.012).

## 4. Discussion

Patients’ perspectives regarding medical students’ involvement in their medical care vary according to different aspects of their previous experiences. Only a little over half of our participants stated that their doctors asked them if the students could be present, and an even smaller percentage said that their doctors had introduced the students correctly. These results were in accordance with those obtained by previous studies [19,20] in which a considerable percentage of patients estimated having had previous experiences with medical students’ participation without being informed or asked for consent [19].

Some studies point out that most clinical teachers, involved in bedside teaching and other clinical settings, do not have formal training in these skills. [21,22] That being said, aside from educating medical students in the humanities and medical ethics, providing clinical teachers with the adequate training and other tools to assure appropriate guidance of their students might be the key to educating more conscious doctors in the future.

Patients interviewed from the Obstetrics/Gynaecology department were the ones who reported the lowest incidence of adequate consent practices being applied and the ones who stated that they would feel more comfortable without the students’ presence. This can be explained by the sensitive nature of the medical speciality itself [5] and because the current process regarding this medical speciality seems to rely on presumed consent instead of an informed one, and the active decision of permitting students’ presence and participation [8,19]. The absence of an adequate consent process can be an obstacle for patients to understand the purpose of the students’ presence and, therefore, lead to their rejection of the students’ presence in the future. Patients’ willingness to accept medical students’ participation in the future might be affected, and no clinical contact in Obstetrics/Gynaecology would result in a critical gap in students’ medical education.

Previous research has shown that receiving the appropriate information and the opportunity to deny students’ participation plays an essential role in patients’ comfort levels and willingness to accept their involvement [19,23]. In accordance, our results show that the department with the lower rate of doctors who asked for patient consent is the one with the higher rate of patients who feel more comfortable without the presence of medical students, and consequently these patients may therefore be less likely to accept students in the future. On the contrary, the department with the higher rate of these practices was also the one with a lower rate of patients who feel more comfortable without the presence of medical students. There is a critical need for clinical teachers and others to ensure adequate measures to inform patients and seek active consent about their involvement in medical teaching [20,24]. Although the adequate number of students present in the room may vary from situation to situation, it seems excessive in any circumstances to have 12 students in a room, as reported by some patients in this study. An adequate number of students for every specific situation should also be sought, and it should be a requirement to avoid an excessive number of people—particularly in more delicate situations, i.e., in a gynaecological examination—assuring patients’ comfort.

Patients’ level of education was also found to have a significant association with the discomfort felt in the presence of students; namely, patients with a higher education were the ones who expressed the most fear of revealing an intimate problem if students were present. Patients with a higher level of education might be more aware of their rights and, if so, might be more offended by students’ presence in the setting of an intimate problem if adequate information about their role was not provided and consent was not requested.

Literature on the matter points out that elderly patients tend to accept students’ participation more often without being informed [19]. However, our study showed that patients who reported having been asked for consent by students were older than those who stated otherwise. In line with this, our results also show that younger patients were the ones who expressed having felt uncomfortable with the students’ presence the most and were the ones who would feel most bothered by students’ participation in case their condition involved private parts of their bodies. Once again, being adequately informed and asked for consent seems to play an essential role in patients’ acceptance, but higher discomfort rates from younger patients might also possibly be explained by a better knowledge of their rights than older ones. Furthermore, a lesser difference in age between patients and the students might be another possible contributor.

Patient gender was also found to be significantly associated with patients’ responses, being that more than half of the male patients stated that they received an adequate explanation and had their doubts clarified regarding the procedures performed by students, while this was only the case for less than a quarter of female patients. Even though some studies suggest that women tend to be more open to bedside teaching [25], the literature also shows that women are not offered the same treatment as men when it comes to healthcare [26]. Despite all efforts to fight gender bias, this result might directly reflect a cultural situation perpetuating gender inequality.

Almost every patient said that they felt satisfied to be contributing to medical students’ education and every single one stated that none of the students were disrespectful on any occasion. However, there are still patients who felt bothered by their presence and patients who would choose not to participate. The explanation as to why that happens might reside in the fact that adequate information was not provided, and proper consent was not obtained.

Interviews were not conducted on patients admitted to the Infectious Diseases department due to the infection risk. Since the same norms apply to medical students, who are not permitted to visit these patients as a part of their clinical training, we estimate that the results were not affected.

Ethics has been an integrated part of the medical curriculum in medical schools worldwide since the beginning of the 20th century; however, given its theoretical nature, it is not easy to define the curricular goals in this field, so that the development and training of communication and interaction skills is crucial. Previous studies have been conducted on skills-oriented educational interventions as a possible way to improve students’ knowledge, skills, and attitudes [27].

Quoting the 1985 DeCamp Report [28], “a medical-ethics curriculum is designed not to improve the moral character of future physicians but to provide those of sound moral character with the intellectual tools and interaction skills to give that moral character its best behavioural expression”.

This study had some limitations, as it was carried out in a single teaching hospital, and only a small number of patients participated in the study. Further multicentric studies with larger populations are needed to confirm the results.

## 5. Conclusions

Given the results of this study, it was possible to verify that the attitude of the medical team remains far from what is expected. Informed consent is an ethical and legal duty for any intervention which aims to protect the patient’s self-determination. The absence of valid informed consent constitutes a violation of good medical practice, and the responsible physician’s disciplinary, civil or criminal liability may be invoked.

In conclusion, even though we are in a time when people are more and more informed and when current medical practice consists of a patient-oriented clinical practice, guided by patients’ wills and needs, the evidence suggests that medical education’s current practice does not always respect for patient autonomy. Patients are an essential part of clinical training, and they might be, or grow to be, less accepting of medical students’ participation due to inadequate behaviour by both students and their teachers.

Ethical practices are fundamentally embedded in medicine and must be adhered to; there is an obvious need to implement measures that provide medical professionals with the tools needed to adequately deal with the ethical dilemmas they face every day. Even though medical ethics is part of the formal medical curriculum, there is a need to intervene and provide medical students with adequate knowledge of ethical values and fair clinical practices which might mean that the teaching of ethics may need to be altered and adapted to medical students’ growing needs in the field. There is also a need to provide them with well-trained clinical teachers who can serve as educators and role models.

## Figures and Tables

**Table 1 ijerph-19-07611-t001:** Sociodemographic and other characteristics of the study population (*n* = 131).

Age Mean (Standard Deviation)	49	(16)
Location *n* (%)		
Consult	80	(61)
Admission	51	(39)
Department *n* (%)		
General Surgery	63	(48)
Infectious Diseases	35	(27)
Obstetrics/Gynecology	33	(25)
Gender *n* (%)		
Male	54	(41)
Female	77	(59)
Civil status *n* (%)		
Single	30	(23)
Married	76	(58)
Widowed	4	(3)
Divorced	11	(8)
Civil union	10	(8)
Education *n* (%)		
1 to 4 years	46	(35)
5 to 12 years	57	(44)
Higher education	28	(21)

**Table 2 ijerph-19-07611-t002:** *n* (%), of answers to the questions regarding the presence of medical students on a previous consultation and/or hospital admission according to the department where the interview took place.

		Department in Which the Interview Took Place			
	Total *n* = 101	General Surgery *n* = 44	Infectious Disease *n* = 27	Obstetrics/ Gynecology *n* = 30	*p*
The doctor asked me if the students could be present.	60 (59)	26 (59)	22 (82)	12 (40)	0.006
The doctor introduced the students with their name and year of formation.	28 (28)	9 (21)	15 (56)	4 (13)	0.001
I felt uncomfortable with the students’ presence.	19 (19)	4 (9)	3 (11)	12 (40)	0.002
The students introduced themselves as medical students and asked for my consent to collect my medical history/perform the medical examination.	51 (51)	22 (50)	20 (74)	9 (30)	0.004
The students explained the procedures they wanted to perform and clarified any doubts I might have had.	45 (45)	15 (34)	22 (82)	8 (27)	<0.001
When there are medical students present I feel like I receive a better explanation about my condition/illness.	20 (20)	12 (27)	5 (19)	3 (10)	0.184
I feel happy to contribute to the medical students’ formation.	100 (99)	43 (98)	27 (100)	30 (100)	0.520
In case the problem was in a more intimate part of my body I would feel more bothered by the medical students’ presence.	41 (41)	15 (34)	7 (26)	19 (63)	0.008
I’m afraid to reveal an intimate problem in the presence of a student.	37 (37)	15 (34)	8 (30)	14 (47)	0.369
If I could choose, I would feel more comfortable without the presence of medical students.	35 (35)	14 (32)	5 (19)	16 (53)	0.019

**Table 3 ijerph-19-07611-t003:** *n* (%), of answers to the questions regarding the presence of medical students on a previous consultation and/or hospital admission according to patients’ level of education.

		Years of School Attendance			
	Total *n* = 101	1 to 4 Years *n* = 44	Between 5 and 12 Years *n* = 27	Higher Educattion *n* = 30	*p*
The doctor asked me if the students could be present.	60 (59)	17 (53)	28 (58)	15 (71)	0.406
The doctor introduced the students with their name and year of formation.	28 (28)	9 (28)	16 (33)	3 (14)	0.266
I felt uncomfortable with the students’ presence.	19 (19)	3 (9)	9 (19)	7 (33)	0.092
The students introduced themselves as medical students and asked for my consent to collect my medical history/perform the medical examination.	51 (51)	20 (63)	23 (48)	8 (38)	0.196
The students explained the procedures they wanted to perform and clarified any doubts I might have had.	46 (45)	12 (38)	23 (48)	10 (47)	0.624
When there are medical students present I feel like I receive a better explanation about my condition/illness.	20 (20)	6 (19)	9 (19)	5 (24)	0.875
I feel happy to contribute to the medical students’ formation.	102 (99)	32 (100)	47 (98)	21 (100)	0.573
In case the problem was in a more intimate part of my body I would feel more bothered by the medical students’ presence.	41 (41)	9 (28)	20 (42)	12 (57)	0.107
I’m afraid to reveal an intimate problem in the presence of a student.	37 (37)	9 (28)	13 (27)	15 (71)	0.001
If I could choose, I would feel more comfortable without the presence of medical students.	35 (35)	11 (34)	15 (31)	9 (43)	0.647

**Table 4 ijerph-19-07611-t004:** Mean (standard deviation) of the age of the inquired patients who answered affirmatively or negatively to the questions about the presence of medical students on a previous consultation and/or hospital admission.

	NO Mean (sd)	YES Mean (sd)	*p*
The doctor asked me if the students could be present.	50 (17)	48 (15)	0.604
The doctor introduced the students with their name and year of education.	48 (16)	52 (14)	0.217
I felt uncomfortable with the students’ presence.	51 (15)	41 (16)	0.006
The students introduced themselves as medical students and asked for my consent to collect my medical history/perform the medical examination.	46 (17)	53 (14)	0.017
The students explained the procedures they wanted to perform and clarified any doubts I might have had.	51 (15)	47 (16)	0.248
When there are medical students present I feel like I receive a better explanation about my condition/illness.	49 (16)	50 (13)	0.758
I feel happy to contribute to the medical students’ formation.	53 (15)	45 (16)	0.014
In case the problem was in a more intimate part of my body I would feel more bothered by the medical students’ presence.	50 (15)	47 (17)	0.363
I’m afraid to reveal an intimate problem in the presence of a student.	50 (14)	47 (19)	0.586
If I could choose, I would feel more comfortable without the presence of medical students.	50 (17)	48 (15)	0.604

sd: standard deviation.

## Data Availability

The corresponding author can obtain the exact data.

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
