# Peer review of "Medical Education: Patients’ Perspectives on Clinical Training and Informed Consent"

_ijerph, 2022, doi:10.3390/ijerph19137611_

Round 1

Reviewer 2 Report

I read the article with great interest and attention and I think it is an innovative and well-constructed starting point. I am not aware that there are other contributions in the international literature which have taken into consideration this peculiar aspect typical of university hospitals but which actually has an importance. I therefore believe that the article should be published, the authors have collected the data with a good methodology however there are some ideas for improvement:

- the bibliography should include a greater number of elements from the international literature regarding informed consent, for example the authors argue that in recent years the importance of self-determination has increased could cite some examples of this for example: doi: 10.1111 / vox.13106. doi: 10.23736 / S0375-9393.18.13179-8. DOI: 10.1016 / j.transci.2020.102823

- the authors could briefly indicate a) the limits of the study in particular of the questionnaire administered eg. Does the situation change in other departments? and b) any suggestions on how to improve this aspect of informed consent in a practical way.

Round 2

Reviewer 1 Report

A very much improved paper which addresses all the points raised in the first review.  Needs a thorough proofread - e.g., the words 'sentience' and 'obliged' are used wrongly at 216 and 305, respectively - hence, 'minor revisions' (small amount of 'text editing') but otherwise, very good.

Author Response

Comment 1:

A very much improved paper which addresses all the points raised in the first review.  Needs a thorough proofread - e.g., the words 'sentience' and 'obliged' are used wrongly at 216 and 305, respectively - hence, 'minor revisions' (small amount of 'text editing') but otherwise, very good.

Response/Actions: Thank you for your comment. Following your suggestion, we have a proofread and we changed some words that are underlined in red.

Reviewer 2 Report

the article appears much improved according to the indications made and I believe it is ready to be published

Author Response

Response/Actions: we appreciate your comments. Thank you.